# Chitosan Shrinking Fibers for Curing-Initiated Stressing to Enhance Concrete Durability [note 1]

**DOI:** 10.3390/ma18071574

**Published:** 2025-03-31

**Authors:** Dryver Huston, Mandar M. Dewoolkar, Diarmuid Gregory, Mohammad Abdul Qader, Bismark Yeboah

**Affiliations:** 1Department of Mechanical Engineering, University of Vermont, Votey Hall, 33 Colchester Ave., Burlington, VT 05405, USA; diarmuid.gregory@gmail.com; 2Department of Civil and Environmental Engineering, University of Vermont, Votey Hall, 33 Colchester Ave., Burlington, VT 05405, USA; mandar.dewoolkar@uvm.edu (M.M.D.); mabdulqa@uvm.edu (M.A.Q.); bismark.yeboah@uvm.edu (B.Y.)

**Keywords:** cracking, early ages, fiber-reinforced concrete, shrinkage, pH activated fiber, durability, freeze-thaw, chloride penetration

## Abstract

Concrete, a widely used construction material, faces environmental concerns and limited durability. This study aimed to enhance concrete durability and long-term resilience by incorporating shrinking chitosan-based fibers. Two types of fibers were used: active fibers, which shrink upon initiation of the curing cycle in concrete, and passive fibers, which served as experimental controls and were pre-shrunk before being added. Two types of durability tests were conducted: freeze-thaw and chloride penetration. Fiber ratios of 0.5 wt%, 1 wt%, and 2 wt%, along with a 0 wt% control, were used for freeze-thaw testing, while ratios of 0.24 wt%, 0.36 wt%, 0.5 wt%, 1 wt%, and 2 wt% were used for chloride penetration. Results demonstrated significant durability improvements with active fibers. Active fiber-reinforced concrete exhibited over 200% greater freeze-thaw durability than passive fibers and over 500% compared to non-reinforced concrete. Chloride penetration testing revealed reduced penetration rates in active fiber concrete. After saltwater exposure, active fiber-reinforced concrete showed up to 59% higher resistance than passive fibers and 249% compared to the control group. These findings highlight the potential of chitosan fibers to significantly enhance concrete durability, paving the way for more sustainable construction practices.

## 1. Introduction

Concrete, a building material with a history spanning millennia, has evolved from rudimentary cements to modern Portland cement-based composites. Today, global concrete consumption exceeds 30 billion tons annually—three times the per capita usage of 40 years ago. Amid the climate crisis, concrete sustainability is paramount, given that cement production contributes approximately 8% of global CO_2_ emissions [1,2]. While concrete possesses high compressive strength (17–28 MPa), its inherent weaknesses include low tensile strength and limited ductility. Concrete consists primarily of cement and aggregates, with its key component, Portland cement, containing hydrated lime that maintains a high pH (approximately 12–13.8) [3]. Reinforcement strategies, including rebars and fibers, are employed to address these weaknesses. Rebars improve tensile strength and ductility, while fibers mitigate crack propagation and enhance post-yield and tensile strength [4]. Additionally, pre-stressing techniques utilizing cables significantly enhance tensile capacity. These reinforcement methods optimize concrete performance across diverse applications.

Enhancing concrete durability is crucial to reducing the frequency of replacements, minimizing maintenance costs, and ultimately decreasing CO_2_ emissions associated with concrete production. Concrete degradation stems from internal factors (e.g., alkali-silica reactions) and external threats (e.g., carbonation, sulfate attack, freeze-thaw cycles, and chloride infiltration). Pre-existing microcracks facilitate the intrusion of these damaging agents, accelerating structural deterioration. Mitigating crack formation or limiting their size is essential to strengthening concrete against internal and external stressors, thereby extending its lifespan and reducing environmental impact [5].

Freeze-thaw damage is a major durability concern, occurring when water infiltrates cracks, freezes, expands, and induces stress within the concrete matrix. This cyclic process, common in regions with fluctuating temperatures, significantly compromises structures like bridges. Concrete’s ability to withstand freeze-thaw cycles is a critical determinant of its longevity. Air-entraining admixtures are commonly used to enhance freeze-thaw resistance by introducing microscopic air bubbles that provide space for freezing water to expand. However, these admixtures often reduce overall concrete strength and increase permeability, making them a trade-off rather than an ideal solution [6].

Another significant durability challenge is chloride attack. Chloride ions—originating from deicing salts, saltwater, and contaminated soil—penetrate concrete and degrade its passivating layer, leading to rebar corrosion and structural deterioration. To simulate real-world exposure conditions, freeze-thaw testing often incorporates saltwater to assess the compounded effects of chloride infiltration and thermal cycling.

Pre-stressed concrete, a method that enhances durability by minimizing tensile cracking, is achieved using steel cables in two primary ways: pre-tensioning (where cables are tensioned before concrete is poured) and post-tensioning (where cables within a duct are tensioned after concrete has hardened and fixed with grout). Structural repair strategies include filling macrocracks with resin or grout and externally reinforcing tension faces with fiber-reinforced polymer strips. Emerging self-healing cement technologies are also being explored to enhance durability and reduce maintenance needs [7,8].

Fiber-reinforced concrete, incorporating steel or polymer fibers, mitigates crack propagation by bridging and restraining cracks. Under stress, these fibers either fracture or pull out, thereby increasing concrete’s toughness, tensile strength, and post-crack compressive strength [9,10]. This reinforcement approach significantly enhances durability [11,12,13]. In this study, we investigate the integration of shrinking chitosan-based fibers into concrete to combine the advantages of fiber reinforcement with the prestressing benefits typically associated with steel cables.

Chitosan, derived from the deacetylation of chitin—a biopolymer found in crustacean shells—is one of the most abundant naturally occurring polymers, with an estimated 10^11^ tons produced annually [14]. This quantity is approximately 300 times greater than the annual production of plastic [15]. Chitosan is widely studied for its antimicrobial properties in medical and food applications and is recognized for its biodegradability [16]. In the context of concrete, chitosan’s unique property of shrinking in high-pH environments, such as fresh cementitious matrices, presents a novel approach to enhancing mechanical performance through internal prestressing.

Chitosan fibers can be fabricated using various techniques, including electrospinning and manual mixing. However, commercially available electrospun chitosan fibers do not exhibit shrinkage, as they are coagulated in ammonia-rich environments that neutralize this property [17,18]. Similarly, gel-spinning—a variation of electrospinning—prevents shrinkage due to ammonia exposure during fiber formation [18,19]. In contrast, manually mixed and dried chitosan fibers, which are subsequently cut, retain their shrinkage capability since they are not treated with ammonia [20].

Previous studies have explored chitosan as a reinforcement material for concrete, demonstrating its ability to improve mechanical properties and microstructural characteristics. For instance, research incorporating chitosan particles into cementitious composites found that a 0.25 wt% dosage slightly increased compressive strength (~4%) while slowing hydration rates compared to control samples. By 28 days, chitosan-enhanced concrete outperformed control specimens in compressive strength, with SEM analysis indicating ongoing microstructural development in chitosan-containing samples, unlike the fully hydrated control samples [21].

In another study, metakaolin-based geopolymer composites reinforced with chitosan powder exhibited significant mechanical enhancements. At 1 wt% chitosan content, compressive and flexural strengths increased by approximately 16% and 33%, respectively, while flexural toughness improved by 83% compared to the control group. However, at 2 wt% chitosan, improvements diminished due to increased porosity and viscosity [22].

Further research utilizing chitosan fibers derived from shrimp shells demonstrated enhanced mechanical performance in concrete. When 0.5 wt% chitosan was incorporated—with or without 1 wt% steel fibers—compressive strength increased by 21% in the chitosan-only group and 13% in the chitosan-steel fiber group. Additionally, chitosan-reinforced concrete exhibited higher compressive strength than mixtures containing superplasticizers, despite a reduction in workability. Indirect tensile strength increased by over 30%, and flexural strength improved beyond levels achieved with superplasticizers, affirming the durability benefits of chitosan inclusion [23].

This study investigates the novel application of chitosan fibers in concrete, specifically examining their impact on long-term durability. To the best of the authors’ knowledge, no prior research has explored the role of shrinking chitosan fibers in enhancing concrete durability. This study aims to bridge that gap by assessing the shrinking effects of chitosan fibers on concrete’s durability, and this paper extends our previous work, which was presented at [24].

We hypothesize that incorporating pH-sensitive, shrinking chitosan microfibers into the concrete matrix induces internal prestressing, effectively reducing pore size and limiting water infiltration. Additionally, these fibers are expected to create synergistic benefits, such as closing microcracks and interlocking aggregate particles, which collectively enhance the material’s resilience. Chitosan’s inherent hydrophilicity further contributes to durability by absorbing infiltrating water, thereby reducing internal voids and mitigating freeze-thaw damage. As the concrete cures, the shrinkage of these fibers is anticipated to constrict microcracks, reinforcing structural integrity and extending the material’s lifespan.

Figure 1 illustrates the proposed hypothesis, depicting how chitosan fibers contribute to reducing internal pore size through shrinkage, ultimately improving concrete’s resistance to degradation.

We used two types of fibers: active fibers, which shrink initially when added to the concrete, and passive fibers, which are pre-shrunk by immersing them in a highly alkaline solution similar to concrete’s alkalinity and then drying them before being added to the concrete. Various ratios of these active and passive fibers, ranging from 0 wt% (control) to 2 wt%, were used. Their effects on the durability of concrete were evaluated through a laboratory testing program, specifically using freeze-thaw resistivity and chloride penetration testing.

## 2. Research Significance

This study introduces chitosan, a biodegradable biopolymer, to develop shrinking fibers that enhance concrete durability. Unlike traditional fibers, chitosan shrinks in high-pH environments, inducing internal prestressing that reduces pore size, limits water infiltration, and strengthens the structure. Inspired by post-tensioned reinforcement, this microstructural approach improves freeze-thaw resistance and chloride penetration. Enhanced durability extends concrete lifespan, reducing maintenance costs and carbon emissions. This breakthrough in fiber-reinforced concrete offers a sustainable, high-performance solution to increase durability while minimizing environmental impact, aligning with global sustainability goals.

## 3. Materials and Methods

### 3.1. Concrete

A commercially available fast-setting concrete mix (QUIKCRETE, Atlanta, GA, USA) was used in this study, sourced from a hardware store. According to the manufacturer, it achieves a compressive strength of 4000 psi at 28 days. This dry mix contains all essential constituents—cement, fine aggregate, and coarse aggregate—ensuring a consistent composition and improving experimental repeatability. Although the exact proportions of each component are not specified on the packaging, using a pre-mixed product minimizes batch-to-batch variability.

To ensure proper aggregate sizing for the molds, the dry mix was sieved using a 6.35 mm (1/4 inch) sieve. This step guaranteed that the maximum aggregate size was smaller than one-fifth of the smallest mold dimension, in accordance with ACI 318-19 [25]. The concrete was then prepared and poured into molds measuring 160 mm × 40 mm × 40 mm.

### 3.2. Chitosan Fibers

The chemistry of chitin and, ultimately, chitosan is why chitosan fibers will shrink in high pH environments like Portland cement concrete [14]. Chitin has an acetyl group (CH_3_CO) connected to a secondary amine group (−NHR). When chitin is deacetylated, the acetyl group is removed, and the secondary amine group becomes a primary amine group (–NH_2_). Chitosan is not soluble in water but is soluble in acids. Dissolving chitosan into an acid has a peripheral effect of turning the primary amine group into an ammonia (–NH_3_^+^) group by reaction with hydrogen ions in the acid. This protonation causes the polymer to grow through ion-ion repulsion. This polymer is now primed to be shrunk when put into a high pH, alkaline environment. When put in a basic environment, the hydrogen ions are pulled out of the polymer, called amine deprotonation, and it shrinks back down now that it is relieved of the repulsive force.

Chitosan powder, supplied by Sigma Aldrich, Saint Louis, MO, USA (Product Number: 419419, CAS Number: 9012-76-4) [26], was utilized in this study. The key properties of the chitosan powder are summarized in Table 1. Chitosan fibers were fabricated following the method proposed by Fernandez and Ingber [20]. The fibers have an ultimate tensile strength of 56.4 ± 2 MPa and a modulus of toughness of 2.29 ± 0.4 J·cm⁻^3^ [20] by mixing 48.5 g of distilled water, 48.5 g of 1 M acetic acid, and 3 g of chitosan powder for 5 min. The resulting gelatinous chitosan solution was spread thinly and evenly on a baking sheet and air-dried in a well-ventilated area for 24–48 h until fully dry. Subsequently, the chitosan sheet was cut into small fibers using scissors. To assess shrinking fiber-reinforced concrete, tests included control specimens prepared without fibers and with passive fibers. The passive fibers were active fibers that pre-shrunk in a high pH CaO solution. This pre-shrinking process rendered the fibers passive in the concrete, not contributing to any shrinkage support. Half of the active fibers were immersed in a CaO solution by adding 0.5 g of CaO powder to 1 kg of distilled water. The pH of the solution was 12–13, similar to the concrete environment. The fibers were immersed in this solution for 12 h and then air-dried for 24–48 h before being added to the concrete. Figure 2 shows the progression of the fiber at the three mentioned stages, first as active fibers, then after immersion in the CaO solution for 12 h, and then after 24 h of drying to become passive fibers.

### 3.3. Initial Testing of Fibers

Chitosan fibers underwent a pre-shrinking process to evaluate their dimensional stability in a high-pH environment. Five fibers, weighing between 0.5 g and 1 g, were initially measured and submerged in a high-pH CaO solution for 30 min, followed by air drying for 24 h. The fibers were then re-submerged, removed, and re-weighed to assess shrinkage. Additionally, five fibers from the same batch later used in concrete—measuring 48–52 mm in length, 2–3 mm in width, and 0.2 mm in thickness, with an aspect ratio of 250—underwent the identical process. Initial measurements were recorded using digital calipers and the fibers were subjected to the same submersion, drying, and re-measuring cycle to evaluate the pre-shrinking effect.

### 3.4. Concrete Specimen Preparation for Freeze-Thaw Testing

Rectangular prism molds (40 mm × 40 mm × 160 mm) were used for casting concrete. Concrete casting involved labeling, applying petroleum jelly, and sieving a 6.35 mm (1/4”) sieve concrete mix. A total of 2.726 kg of sieved dry concrete mix was combined with an appropriate amount of fibers, followed by the addition of 420 mL (for passive fibers) or 450 mL (for active fibers) of water. Hand-mixing occurred in a bowl using a scoop. Wet concrete was vibrated in molds for 30 s to eliminate air bubbles and the surface was screeded with a trowel. Specimens cured in molds at room temperature and humidity for one day, followed by a 13-day cure cycle in laboratory conditions. Notably, a high humidity curing room was omitted to maximize moisture extraction for complete fiber shrinkage. The weight ratios of fibers used were 0% control, 0.12%, 0.24%, 0.36%, 0.5%, 1%, and 2%, equivalent to fiber weight to concrete volume ratios of 0 kg/m^3^, 2.5 k/gm^3^, 5 kg/m^3^, 7.5 kg/m^3^, 10.4 kg/m^3^, 20.8 kg/m^3^, and 40.7 kg/m^3^, respectively. For each combination of weight ratio and fiber type (active/passive), as well as the control group with 0% fibers, five specimens were tested. Due to limited space in the freeze-thaw chamber, the total number of specimens was divided into two groups of 35 specimens each. The first group comprised the four lower weight ratios, with the 0.24 wt% specimens demonstrating the best performance in the first testing batch. Consequently, an additional five specimens with the 0.24 wt% ratio were prepared and tested alongside the remaining specimens in the second batch, which included higher weight ratios. Table 2 shows the sample description of the specimens.

### 3.5. Chloride Penetration Testing

Three fiber weight ratios—0.24 wt%, 0.5 wt%, and 1 wt%—along with a control group containing 0 wt% were utilized in this test series. Concrete specimens were prepared following the procedure outlined for the freeze-thaw test, with five specimens per group. Each specimen incorporated four 6/32, 1.5-inch stainless steel cap head hex bolts, inserted halfway into the wet concrete to serve as electrodes for resistivity measurements, as illustrated in Figure 3.

The bolts were evenly spaced at 32 mm intervals, exceeding 1.5 times the maximum aggregate size [27], to ensure accurate and consistent measurements. Prior to demolding, a stainless steel hex nut was tightened onto each bolt at the concrete surface to secure it and prevent displacement due to potential impacts. For resistance measurements, an additional stainless steel hex nut and a stainless wingnut were affixed to each bolt to ensure stable and reliable connections.

### 3.6. Freeze-Thaw Fatigue and Testing

Freeze-thaw failure has many theories, including ice crystal formation, hydrostatic pressure hypothesis, and osmotic pressure hypothesis, Litvan freeze-thaw damage theory, critical saturated degree theory, among others. The most prominent of these theories are the hydrostatic pressure hypothesis and the osmotic pressure hypothesis [28,29,30]. When water freezes, it expands by about 9% [31], exerting pressure in concrete pores. This pressure cracks the cement and bridges into new pores, progressively breaking the concrete upon freeze-thaw cycling. Adequate water supply facilitates this process [32], while air-filled pores mitigate damage by limiting water expansion. In freeze-thaw testing, a standard evaluation involves non-destructive resonance tests measuring the dynamic modulus. Dynamic modulus, representing the stress-to-strain ratio under vibratory conditions, is simplified by measuring only the fundamental frequency relative to the initial specimen value. This approach eliminates the need for precise dimensional measurements, allowing comparison of slightly different-sized prisms. Resonance frequency measurements, employing forced or impact resonance, aid efficient testing. The calculation of relative dynamic modulus in percent is based on Equation (1):(1)$$P_{c}=\left(n_{1}^{2}/n^{2}\right)\;\cdot \;100$$
where:

P_c_ = percent relative dynamic modulus of elasticity after N cycles

n = fundamental transverse frequency at 0 cycles

n_1_ = fundamental transverse frequency after N cycles

P_c_ can then be used to calculate the durability factor in Equation (2):(2)$$DF=\frac{P\cdot \;N}{M}$$
where:

DF = durability factor of the specimen

P = percent relative dynamic modulus at N cycles

N = number of cycles when P reaches the minimum value for the test

The minimum P value is typically 50% and the standard number of cycles, M, is generally 300. M represents the point at which a specimen is deemed freeze-thaw resistant. If a specimen can withstand 300 cycles without its P value dropping to 50%, it is considered freeze-thaw resistant, and N is equal to M.

Length change in percent is calculated as follows in Equation (3):(3)$$L_{c}=\frac{l_{1}-l}{l}\;\cdot \;100$$
where:

Lc = percent length change after N cycles

l1 = specimen length after N cycles

l = specimen length at 0 cycles

Similarly, mass change in percent is calculated in Equation (4):(4)$$M_{c}=\frac{m_{1}-m}{m}\;\cdot 100$$
where:

M_c_ = percent length change after N cycles

m_1_ = specimen length after N cycles

m = specimen length at 0 cycles

The freeze-thaw testing generally followed ASTM C666 Procedure A with some modifications. Concrete specimens were wrapped in wet cloth and placed in an environmental chamber cycling between −20 °C and 20 °C. Impact resonance tests are conducted every 14 cycles, each lasting 3 h. The temperature cycle takes 30 min to drop to −20 °C, where it stays for 1 h, then takes 30 min to rise to 20 °C, where it stays for another hour. This 3-h cycle was repeated 14 times and if the testing lasted 294 cycles, the specimens undergo six more cycles before reaching a total of 300 cycles. The test procedure involves the following steps: prisms are removed from the chamber, and their mass and length are recorded. Each prism is dried and a PCB 352A24 accelerometer is connected to a PCB 482A04 power supply, linked to an HP 3566A spectrum analyzer. The analyzer is connected to a Micron Millenia CPU, which interfaces with a monitor. Data are read from the spectrum analyzer via HP 3566A software (https://nscainc.com/wp-content/uploads/pdf/A_3566A.pdf?srsltid=AfmBOooAbYWb6wxLxrtvMuFEdwnYhr43rbc6MEPJ0wfuKumMrlxdwu7G (accessed on 25 February 2025)). A prism, placed on bungee cord supports, has the accelerometer affixed and is struck with a bronze pipe. This process, including data averaging, is repeated for each prism. Afterward, wet cloths are reapplied and the prisms undergo an additional 14 cycles in the environmental chamber.

### 3.7. Resistivity Testing

In this study, the four-point Wenner probe test was used for resistivity measurement. The test employs Kelvin four-terminal sensing, utilizing four equally spaced electrodes to measure concrete resistivity. The outer electrodes generate current through the concrete, while the inner electrodes gauge the voltage drop between them. Knowledge of the applied current and electrode spacing, measured as the distance “a” between two electrodes from center to center, enables the calculation of resistivity using Equation (5):(5)$$\rho =2\pi a\frac{V}{I}$$
where:

ρ = resistivity

a = distance between consecutive electrodes

V = voltage measured by inner electrodes

I = current supplied by outer electrodes

The specimens were cured in their molds for 7 days at room temperature and humidity, after which they were demolded and the resistivity was measured using the above-mentioned four-point Wenner method. An MCT-Li-66 scale and a Stanford Research Systems SR715 LCR Meter, Sunnyvale, CA, USA (10 kHz frequency, 1.0 V drive voltage) were utilized. The setup included the SRS BNC adapter set and 4 BNC to spade lug cables. Electrodes, positioned on a fixture for resistance testing, had a total of four equally spaced electrodes, outer electrodes for driving constant current, and inner electrodes for measuring the potential difference. Specimens underwent wet and dry cycles, involving partial immersion in 3.5% NaCl for 2 days, followed by 5 days of drying, and these two cycles were repeated another eight times. During the wet cycles, the specimens were placed on 2 mm wide supports to allow absorption through their bottom surfaces. Measurements for resistivity and mass were taken at 0, 2, 7, 9, 14, 16, 21, 23, 28, and 30 days.

### 3.8. SEM Images

Following the completion of resistivity testing on various dates, a single specimen from each group underwent scanning electron microscopy (SEM) using the Zeiss Sigma 300 field-emission scanning electron microscope, Oberkochen, Germany. This instrument offers magnifications ranging from 10× to 1,000,000× [33]. The specimens were systematically scanned at various magnifications, specifically 25×, 100×, 250×, 500×, 1000×, 2500×, 5000×, 10,000×, and 20,000×. The primary objective of this analysis was to determine both the entropy and the solid area fraction within the microstructure.

Entropy, a statistical measure of randomness that characterizes the texture of a grayscale image, was calculated using Equation (6):(6)$$Entropy=-\sum \left(p\cdot \log_{2}\left(p\right)\right)$$
where p represents the normalized histogram counts obtained from the image histogram. Higher entropy values indicate greater randomness, corresponding to a higher packing density within the material, the entropy of each image was calculated using MATLAB R2023a code.

The solid area fraction for each group was calculated using images captured at magnifications of 2500×, 5000×, 10,000×, and 20,000×, as these levels provided clear visualization of voids, identifiable as black regions. The initial step involved converting these images into binary format using MATLAB’s image processing tools. After conversion, the area occupied by white pixels—representing solid regions—was determined. Simultaneously, black pixels were used to identify void areas within each specimen.

It is important to note that no special treatment was applied during sample preparation. A small piece of concrete was sectioned from the first specimen of each group following the chloride penetration test, and this sample was used for SEM imaging. This approach ensured that the analysis reflected the natural condition of the material post-exposure. An example of a sample used for SEM analysis is shown in Figure 4.

## 4. Results and Discussion

### 4.1. Fiber Shrinking and Alkaline Solution Absorption

Figure 5 illustrates the relative changes in the length and width of fibers when first immersed in a CaO solution with a high pH, similar to that of a concrete environment. This process is then repeated. The blue solid line represents how active fibers change due to the high pH of the concrete, while the orange dotted line shows how passive fibers change when exposed to the high pH environment for the second time—first when soaked in the CaO solution for 30 min and then when added to concrete.

Figure 5 also shows that both types of fibers undergo very similar changes in length and width. Table 3 confirms this by showing the absorption of the CaO solution in grams, with only minor differences between the fibers. These minor differences eliminate variables such as size, shape, elastic modulus, tensile strength, and pull-out force. Thus, the intensity to shrink remains the only difference between active and passive fibers.

### 4.2. Mass Results of Freeze-Thaw Testing

As seen in Figure 6, there is an initial increase in mass of concrete for all groups after the first 14 freeze-thaw cycles. The results also show that mass gain was higher for both higher wt% ratios and for the active fiber groups compared to their corresponding passive fibers. Even after the first 28 cycles, the higher wt% specimens continued to gain mass. For 1 wt% and 2 wt% active fibers, this mass gain lasted up to 150 cycles, likely due to the fibers absorbing from the wet cloths they were in during testing. The 2 wt% passive fibers exhibited similar behavior but failed after 154 cycles.

The recorded increase in mass is due to the dry curing of the specimens, which were not covered with a wet cloth or kept in a humid environment but were cured in air. The dry curing ensured that the chitosan fibers would shrink in the concrete, as they only shrink when dry. When the samples were wrapped in wet cloths for freeze-thaw cycling, they absorbed water from the cloths, leading to increased mass. Some groups even continued to gain mass after 28 freeze-thaw cycles by soaking up more water from the cloths. However, after this point, the specimens began to lose mass due to spalling. The freeze-thaw degradation caused pieces on the outside of the concrete to spall off. Resistance to this breaking is a favorable characteristic and appears to correlate directly with the relative dynamic modulus.

### 4.3. Relative Dynamic Modulus

The impact resonance testing reveals a frequency spectrum for each specimen. The fundamental frequency is manually identified by selecting the lowest distinct peak or spike, excluding low-frequency vibrations caused by bungee supports. This determination becomes challenging as the concrete breaks down. The identified frequency at 0 cycles is taken as “n”, the fundamental transverse frequency, used to calculate the relative dynamic modulus for a specimen throughout subsequent tests. Each round of impact resonance testing results in the average relative dynamic modulus for all specimens in a fiber group. Figure 7 displays the averaged relative dynamic moduli for each fiber group and test. Each fiber weight ratio includes active and passive groups, along with a 0 wt% control group. Following initial testing of lower weight ratios, the 0.24 wt% active fiber group excelled. This result prompted a repeated set of testing with the 0.24 wt% active fiber mix for verification and denoted in the results as 0.24 wt% (round 2). Figure 8 reveals three top-performing groups: 0.24 wt% active (round 2), 1 wt% active, and 2 wt% active. While none survived the full 300 cycles, they reached 252, 280, and 238 cycles, exhibiting notable improvement over corresponding passive fiber groups and the 0 wt% control. This improvement aligns with the hypothesis mentioned earlier in Figure 1. The durability factors are reported in Figure 9, which highlight four active fiber groups surpassing their passive counterparts. Notably, at 0.24 wt% (round 1), 0.36 wt%, and 0.5 wt%, active and passive groups exhibit similarity. Significant disparities between 1 wt% and 2 wt% active/passive groups reinforce confidence in the durability-enhancing effect of shrinking fibers at those weight ratios. Overall, except for 0.12, 1, and 2 wt% passive groups, all fiber groups demonstrated increased durability compared to the 0 wt% control.

### 4.4. Mass Results of Chloride Penetration Test

In Figure 10, Figure 11 and Figure 12, a substantial mass increase is observed after the first wet cycle, attributed to the prisms absorbing moisture from the partially immersed saltwater. Following the second dry cycle, an anticipated decrease in mass compared to the first wet cycle is apparent, although the masses remain higher than those recorded after the first dry cycle. This trend persists across all four cycles; after a dry cycle, there is a decrease in the mass of all specimens, followed by an increase after a wet cycle. Figure 10, Figure 11 and Figure 12 illustrate that in all three different fiber weight categories the mass gain for the active group surpasses that of the passive and control groups. This outcome was expected, considering that active fibers absorb more water than passive fibers, which undergo pre-shrinking before being added to the concrete, and the control group that lacks any fibers.

### 4.5. Electrical Resistivity

Figure 13, Figure 14 and Figure 15 present the resistivity outcomes for the three different fiber weights across all testing dates. Notably, resistance is highest at the first dry cycle and lowest at wet cycles surpassing their respective dry cycles. Active fiber groups generally exhibit improved resistivity after each dry-wet cycle, except for the 1% fiber group; this deviation is observed due to a 21.9% decrease in resistivity on the fourth wet cycle compared to the previous third wet cycle. The overall resistivity change for active fiber groups between the last and first wet cycle is 48.5%, 67.7%, and 40.5% for the 0.24%, 0.5%, and 1% fiber weight groups, respectively. The increased resistivity observed in active fiber groups is hypothesized to be due to the post-stressing they apply to the concrete after each wet cycle. This process reduces pore size, limits water absorption—the main factor increasing resistivity—and thereby decreases electrical resistivity. This is supported by chloride penetration results here, which show that the resistivity of both active and passive fiber groups is higher than that of their control specimens, aligning with the hypothesis mentioned earlier in Figure 1. Only the 0.5% passive fiber group shows an improvement between the last and first wet cycle, with a 39.3% increase. Comparing the change between the last and first dry cycle for different fiber weight groups reveals a more significant decrease in resistivity for passive groups. The active fiber groups exhibit a decrease in resistivity between the first and last dry cycle of −87.5%, −85%, and −80.2% for 0.24%, 0.5%, and 1%, respectively, while the passive fiber groups show decreases of 98.2%, 95%, and 98.5% for the same weight ratios. This suggests that passive groups experience higher chloride penetration compared to active groups. On average, active fiber groups demonstrate an 87%, 34%, and 56% higher resistivity than their respective passive fiber groups for 0.24%, 0.5%, and 1% weight ratios. Additionally, active fiber groups exhibit a 249% higher resistivity than the control group.

### 4.6. SEM Image Analysis Results

Figure 16 shows an example for the SEM images for one of the samples an the figure shows the scanning of the same sample at different magnifications. Figure 17 presents the average entropy of images for each group, revealing that the 0.5% active group has the highest entropy, followed by the 0.24% passive group. In contrast, both the 0.24% active and control groups exhibit the lowest entropy values. This suggests that all fiber groups, except the 0.24% active group, demonstrate higher entropy than the control group. These findings are consistent with the dynamic modulus results, indicating the control group’s early failure compared to all fiber groups. Figure 18 shows the solid region area for different groups. Active fiber groups exhibit a significantly higher area, implying a more densely packed structure and supporting the notion that fibers are prestressing the concrete. The 0.24% active group shows the highest area, followed by the 1% active group, which also displayed high entropy. In contrast, the 0.24% passive group and the control group have the lowest area compared to other fiber groups. These results emphasize the positive impact of fiber groups on concrete, particularly the active fibers, suggesting enhanced durability. The key takeaway from the SEM image analysis is that both entropy and solid region area outcomes collectively indicate values that correlate with superior performance for both fiber groups compared to the 0% control group, with the 0.24% weight active mix having the lowest entropy and highest solid area fraction. Despite individual variations within fiber groups, the overall trend suggests that the incorporation of fibers positively influences the concrete’s characteristics, as evidenced by lower entropy values and increased solid region area.

## 5. Conclusions

Two types of tests were conducted on chitosan fiber-reinforced concrete to assess its durability, revealing promising results. Active shrinking fibers exhibited increased freeze-thaw durability at specific weight ratios, showcasing their potential for improving resilience in harsh conditions. Moreover, these active fibers demonstrated superior resistance to chloride penetration compared to passive fiber reinforcement. The unique attributes of chitosan fibers, including automatic shrinking, biodegradability, and widespread availability, position them as promising candidates for widespread use in various applications. The study’s findings are summarized as follows:The mixes with higher fiber ratios, i.e.,1 wt% and 2 wt% active shrinking fiber groups, showed higher durability factors than their respective passive groups in freeze-thaw testing. The 2 wt% active group exhibited a 219% greater durability factor than the 2 wt% passive group. The relatively sparse 0.12 wt% and 0.24 wt% fiber mixes also showed superior durability factors, while the intermediate 0.35 36 wt% and 0.5 wt% exhibited inferior durability.After 300 freeze-thaw cycles, only the 2 wt% active group retained all specimens intact, while the 1 wt% and 0.24 wt% active groups-maintained mass well but experienced prism breakage after 250 cycles, in contrast, the 0 wt% control group failed faster (less than 50 cycles) in terms of length and relative dynamic modulus compared to the active and passive fiber.Active fiber groups absorbed more mass than passive groups during wet cycles but maintained it after dry cycles, demonstrating higher resistivity despite increased mass gain in resistivity testing. This resilience is attributed to post-stressing applied to the concrete following each wet cycle, which effectively reduces pore size, limits water absorption, and increases electrical resistivity.Initial testing on day 0 revealed higher resistance values for passive groups than active groups, except for the 0.24 wt% groups. Over subsequent dry-wet cycles, active fiber groups consistently exhibited improved resistivity, further enhanced by the post-stressing treatment. On average, active fiber groups demonstrated 59% and 249% compared to their respective passive and control groups.SEM image analysis of the results aligns with dynamic modulus findings and electrical resistivity results reinforcing the beneficial impact of fiber reinforcement on concrete properties. The higher entropy and solid region area observed in fiber-reinforced groups further support their enhanced performance compared to the control group without fibers.

An overall conclusion from these studies is that chitosan shrinking fibers have the potential to alter concrete during the curing process in a manner that positively affects durability.

## Figures and Tables

**Figure 1 materials-18-01574-f001:**
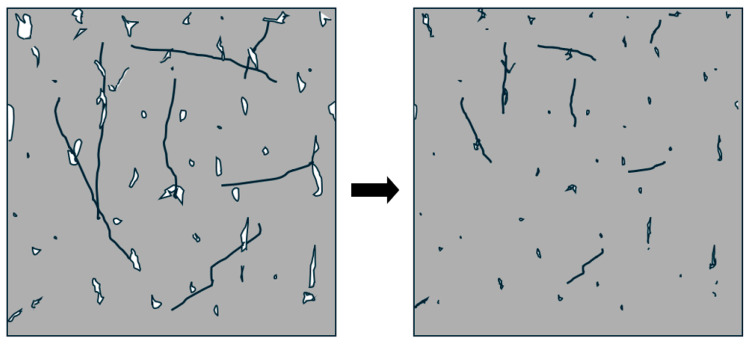
Hypothesis of a stronger composite with chitosan fibers. Voids and fibers are shown before shrinkage (**left**) and after fiber activation, where they shrink, stressing the structure and reducing void size (**right**).

**Figure 2 materials-18-01574-f002:**
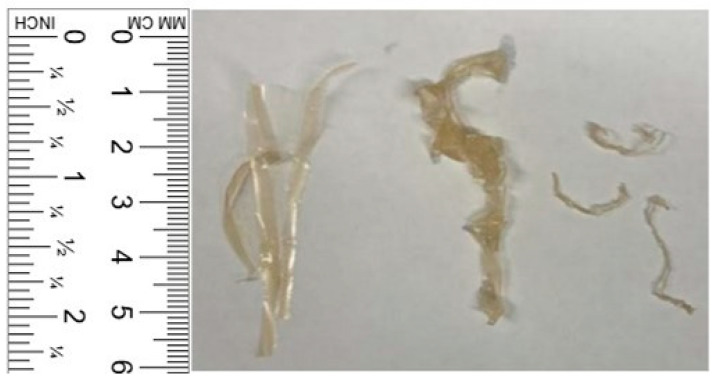
The chitosan fiber is presented in three distinct stages: first, as an active fiber immediately after being cut by scissors (**left**); next, following 12 h of immersion in a high pH CaO solution (**middle**); and finally, after 24 h of drying, where it transitions into a passive fiber for use as a control (**right**).

**Figure 3 materials-18-01574-f003:**
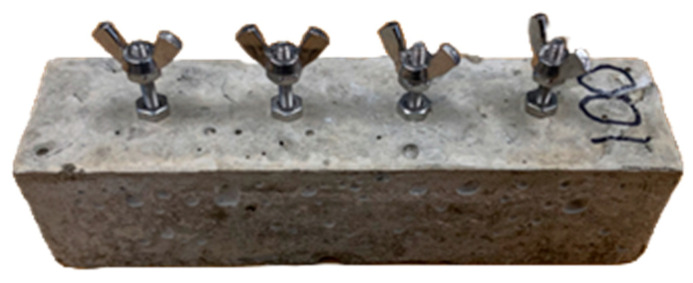
Concrete specimen (40 mm × 40 mm × 160 mm) with electrode bolts and spade lug clamping nuts.

**Figure 4 materials-18-01574-f004:**
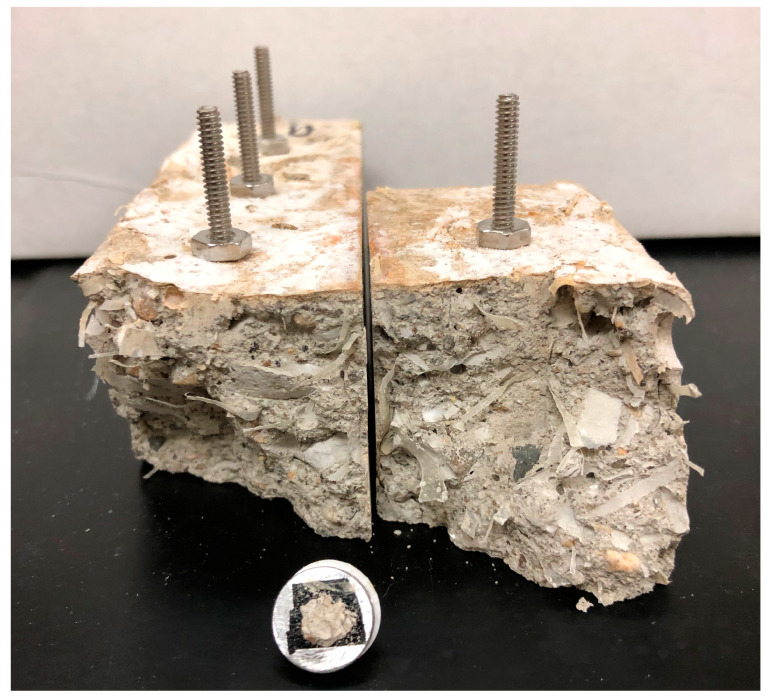
Example of a sample used for SEM image analysis. The sample was sectioned from one of the concrete specimens used for chloride penetration testing.

**Figure 5 materials-18-01574-f005:**
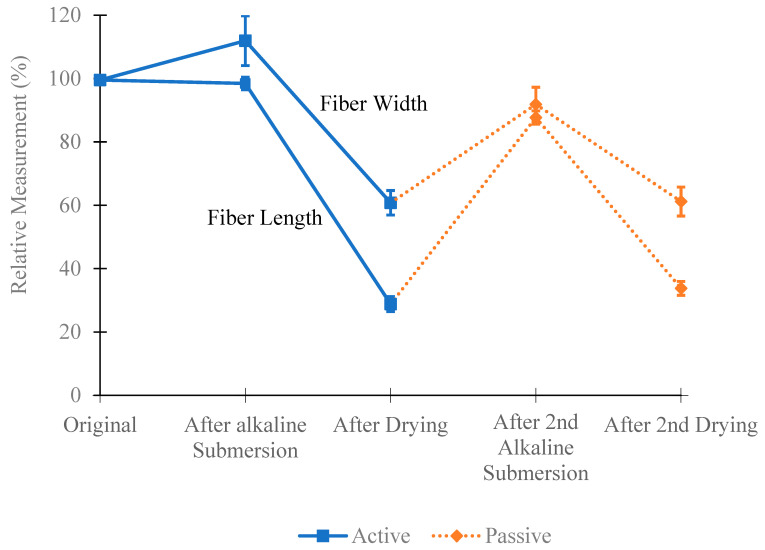
Average measurements of fiber length and width over double alkaline exposure with standard error of the mean error bars.

**Figure 6 materials-18-01574-f006:**
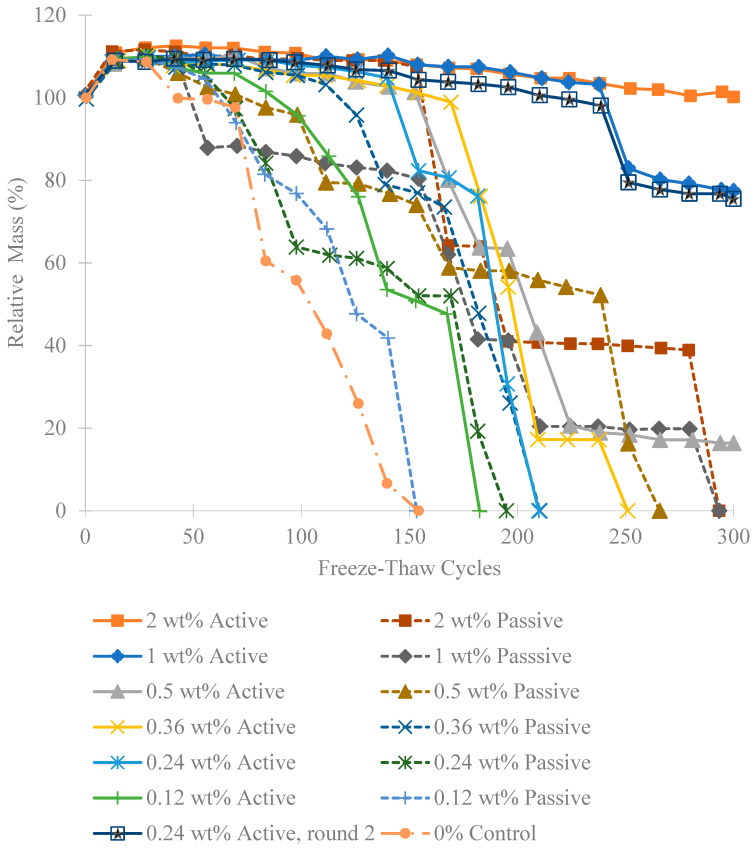
Relative mass of each group over freeze-thaw cycles.

**Figure 7 materials-18-01574-f007:**
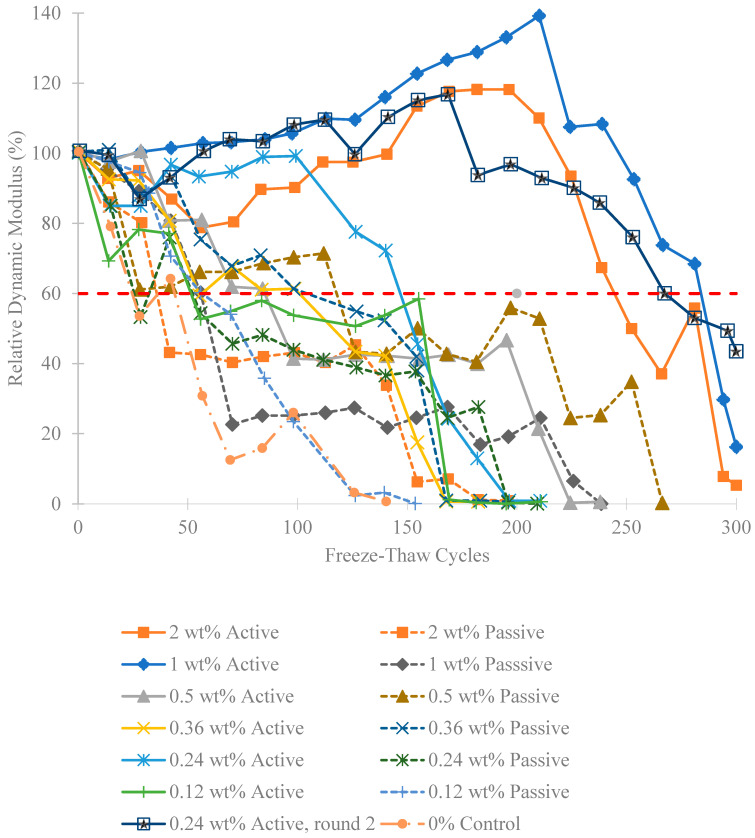
Relative dynamic modulus plot of various fiber wt% over the number of freeze-thaw cycles.

**Figure 8 materials-18-01574-f008:**
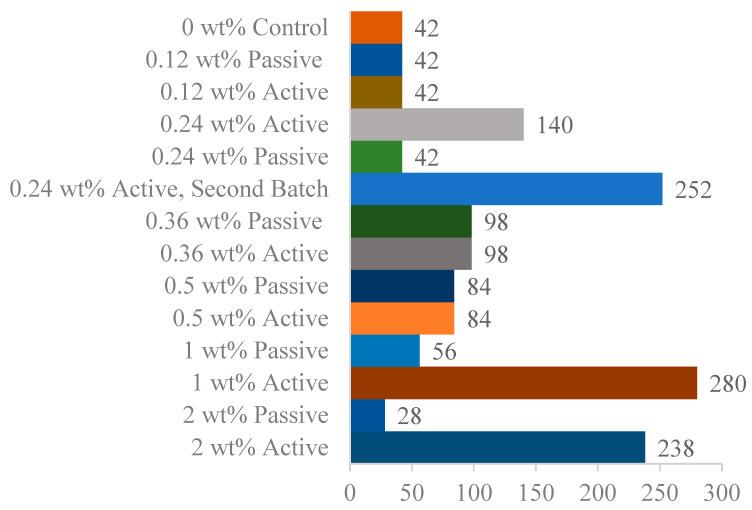
The number of freeze-thaw cycles at which specimens in each fiber group fail, indicated by a dynamic modulus drop below 60% of its original value at zero cycles.

**Figure 9 materials-18-01574-f009:**
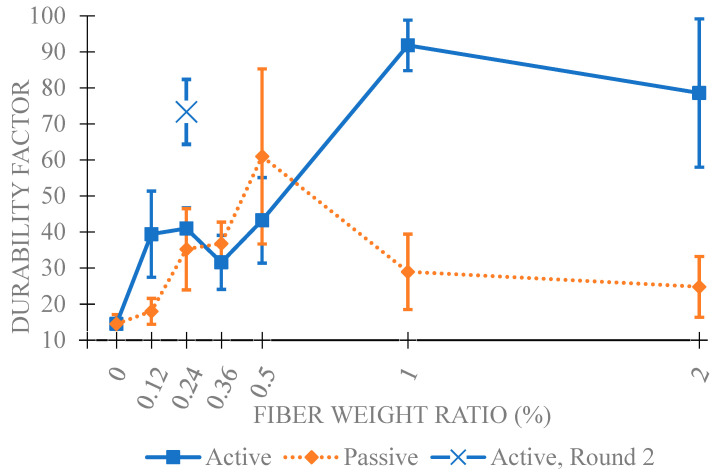
Durability factor plot of various fiber wt% after freeze-thaw cycle failure with standard error of mean bars.

**Figure 10 materials-18-01574-f010:**
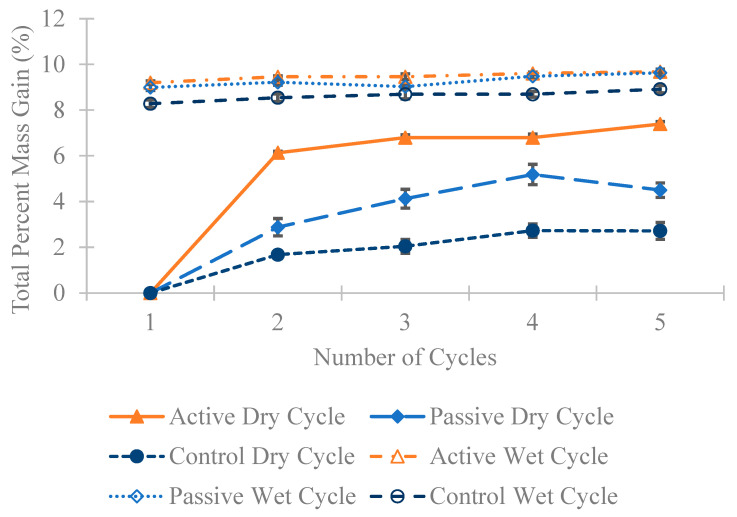
The average percentage increase in mass for the 0.24 wt% fiber group following multiple dry and wet cycles in comparison to their initial mass with standard error of mean bars.

**Figure 11 materials-18-01574-f011:**
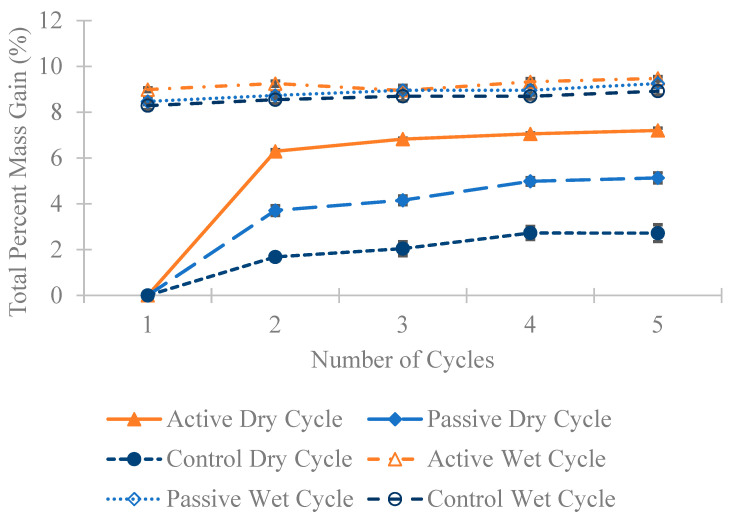
The average percentage increase in mass for the 0.5 wt% fiber group following multiple dry and wet cycles in comparison to their initial mass with standard error of mean bars.

**Figure 12 materials-18-01574-f012:**
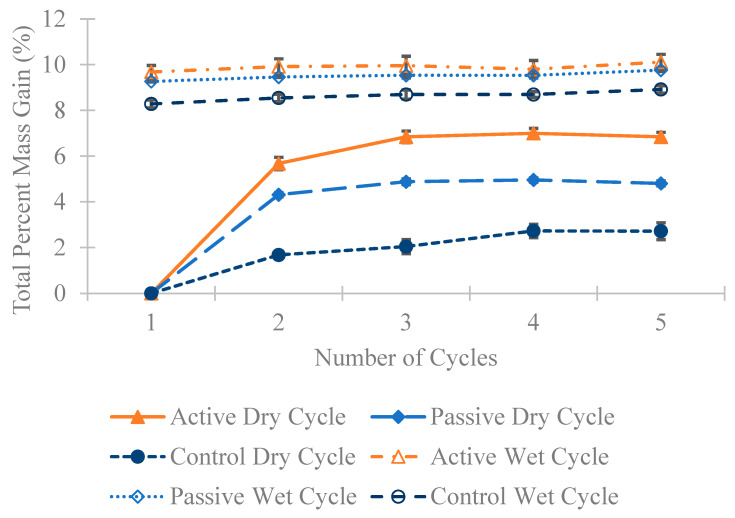
The average percentage increase in mass for the 1 wt% fiber group following multiple dry and wet cycles in comparison to their initial mass with standard error of mean bars.

**Figure 13 materials-18-01574-f013:**
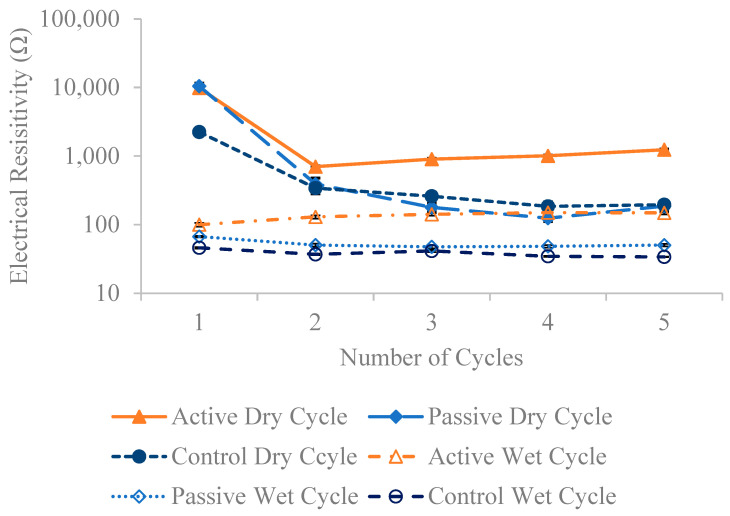
The average electrical resistivity for 0.24 wt% fiber group following multiple dry and wet cycles in comparison to their initial resistivity with standard error of mean bars.

**Figure 14 materials-18-01574-f014:**
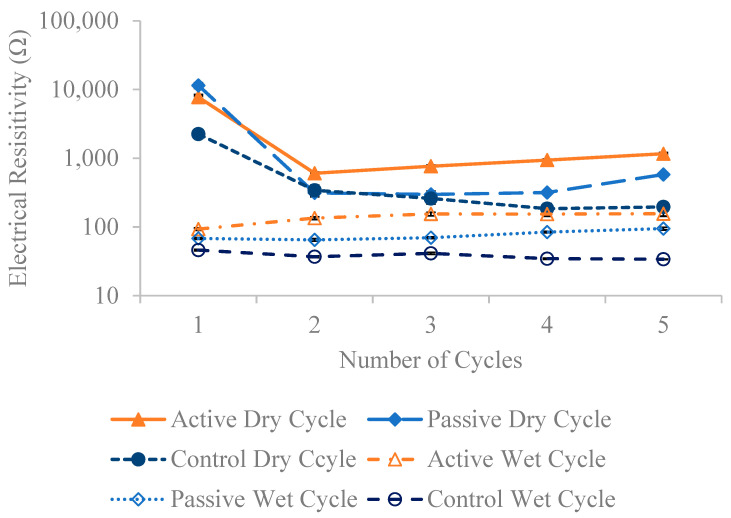
The average electrical resistivity for the 0.5 wt% fiber group following multiple dry and wet cycles in comparison to their initial resistivity with standard error of mean bars.

**Figure 15 materials-18-01574-f015:**
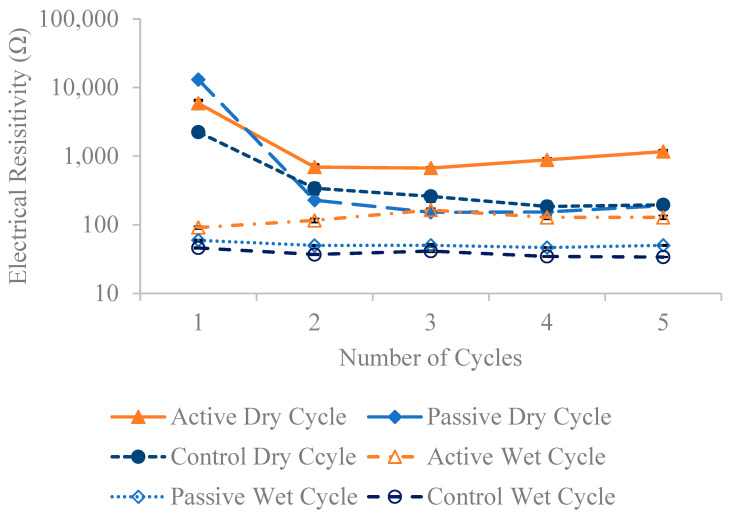
The average electrical resistivity for 1 wt% fiber group following multiple dry and wet cycles in comparison to their initial resistivity with standard error of mean bars.

**Figure 16 materials-18-01574-f016:**
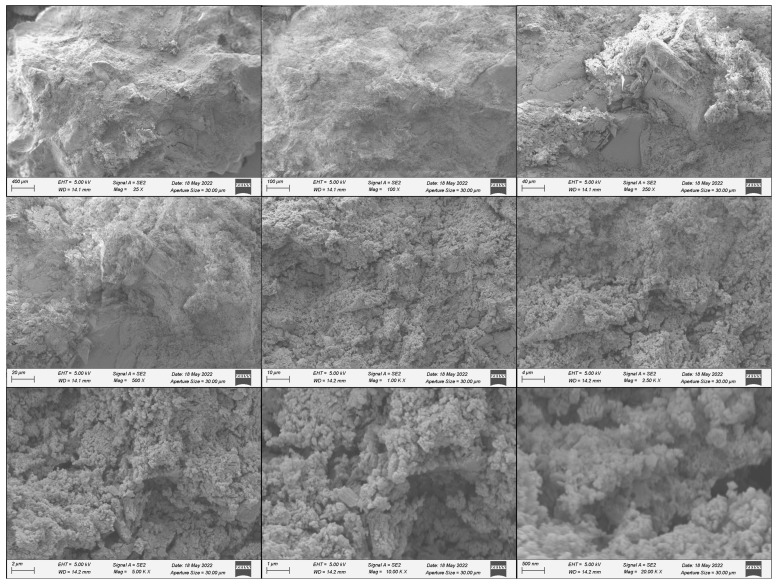
SEM images of the 0.5 wt% passive chitosan concrete specimen are presented at varying magnifications of 25×, 100×, 250×, 500×, 1000×, 2500×, 5000×, 10,000×, and 20,000×, arranged clockwise from the top-left corner.

**Figure 17 materials-18-01574-f017:**
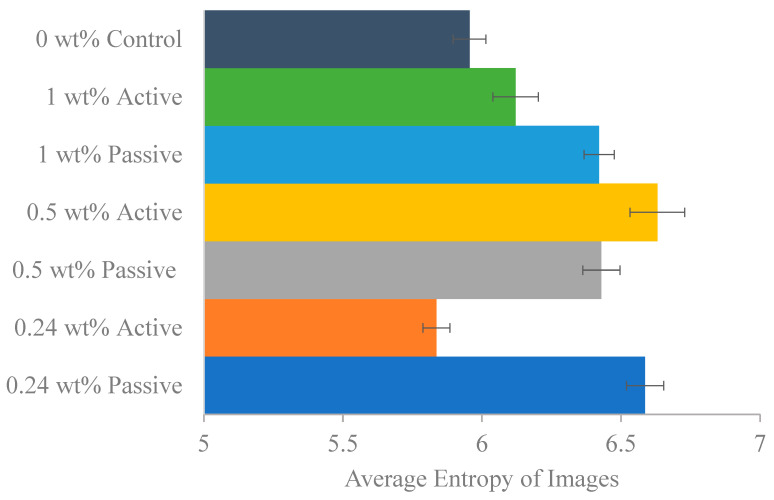
The mean entropy of SEM images for each fiber group with standard error of mean bars.

**Figure 18 materials-18-01574-f018:**
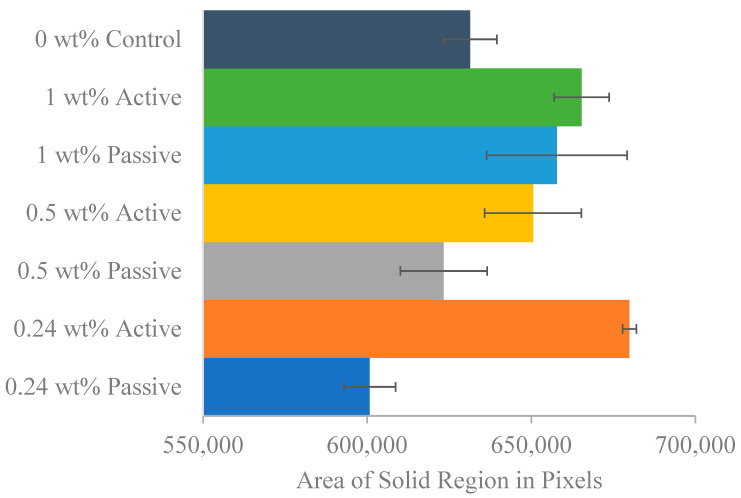
The mean solid region area of binary images for each fiber group with standard error of mean bars.

**Table 1 materials-18-01574-t001:** Properties of chitosan powder [26].

Appearance	Off-White/Beige Powder
Color after acidic	Faint yellow bulk
Solution density	0.15–0.3 g/cm^3^
Deacetylate rate	≥75%
Molecular weight	387 kg/mol

**Table 2 materials-18-01574-t002:** Specimen designation: sample numbers, test types, fiber ratios, and types.

Specimen Number	Test	Fiber Type
1–5	Freeze-thaw	0.12 wt% passive
6–10	Freeze-thaw	0.12 wt% active
11–15	Freeze-thaw	0.24 wt% passive
16–20	Freeze-thaw	0.24 wt% active
21–25	Freeze-thaw	0.36 wt% passive
26–30	Freeze-thaw	0.36 wt% active
31–35	Freeze-thaw	0 wt% control
36–40	Freeze-thaw	0.5 wt% passive
41–45	Freeze-thaw	0.5 wt% active
46–50	Freeze-thaw	1 wt% passive
51–55	Freeze-thaw	1 wt% active
56–60	Freeze-thaw	2 wt% passive
61–65	Freeze-thaw	2 wt% active
66–70	Freeze-thaw	0.24 wt% active round 2
71–75	Chloride penetration	0.24 wt% passive
76–80	Chloride penetration	0.24 wt% active
81–85	Chloride penetration	0.5 wt% passive
86–90	Chloride penetration	0.5 wt% active
91–95	Chloride penetration	1 wt% passive
96–100	Chloride penetration	1 wt% active
101–105	Chloride penetration	0 wt% control
71	SEM analysis	0.24 wt% passive
76	SEM analysis	0.24 wt% active
81	SEM analysis	0.5 wt% passive
86	SEM analysis	0.5 wt% active
91	SEM analysis	1 wt% passive
96	SEM analysis	1 wt% active
101	SEM analysis	0 wt% control

**Table 3 materials-18-01574-t003:** Alkaline solution absorption of active and passive chitosan fibers.

	Passive Absorption (g/g)	Active Absorption (g/g)
Average	3.866	3.870
SEM	0.106	0.158

## Data Availability

The original contributions presented in this study are included in the article. Further inquiries can be directed to the corresponding author.

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
