# Peer review of "Chitosan Shrinking Fibers for Curing-Initiated Stressing to Enhance Concrete Durabilityâ€"

_materials, 2025, doi:10.3390/ma18071574_

Round 1
Reviewer 1 Report
Comments and Suggestions for Authors
The manuscript entitled “Chitosan Shrinking Fibers for Curing-Initiated-Stressing to Enhance Concrete Durability” is partly in line with the Materials journal. This article is based on original research. The topic is up-to-date. This manuscript has proper composition, but before publication, it requires slight revision and clarification, as follows:
- Line 13: remove double point
- Line 35 and others: use down index in chemical formula
- Chapter 2. The novelty of the research is not presented.
- Chapter 3.1. What type of cement was applied (class)?
- Chapter 3.1. What kind of aggregate was applied (minerals / rock)?
- Table 1. Capital letter in line 3
- Chapter 3.2. What was the average size of the fibres and standard deviation?
- Chapter 3. What about statistical analysis? How many samples were tested in each research?
- Chapter 3. Add a table with sample designation. Explain why different mixtures were used for the presented research.
- Chapter 3.8. How were the samples prepared?
- Chapter 4.6. Add some figures presenting the SEM images. What was exactly visible under very high resolutions such as 20,000?
Author Response
Response to Reviewers' Feedback
We sincerely thank the reviewers for their time, thoughtful insights, and valuable suggestions. Their feedback has been crucial in strengthening the manuscript and enhancing its potential for publication. Below is a detailed, point-by-point response to each of the three reviews.
Reviewer 1:
• Line 13: Remove double point
The double point has been removed.
• Line 35 and others: Use subscript in chemical formulas
All chemical formulas have been reviewed and corrected to use appropriate subscripts.
• Chapter 2: The novelty of the research is not presented
This section has been rewritten to clearly present the novelty of the research. The primary innovation is the assessment of self-shrinking fibers that can be chemically activated by cement reactions, focusing on their impact on concrete durability. Specifically, two durability aspects were evaluated: freeze-thaw resistance and chloride penetration. The results demonstrated that chitosan fibers enhance these properties, suggesting their potential for broader application in concrete. It is emphasized that this study is preliminary, and the findings aim to encourage further global research into the self-healing effects of such fibers in concrete.
• Chapter 3.1: What type of cement was applied (class)?
Chapter 3.1: What kind of aggregate was applied (minerals/rock)?
The research utilized a commercially available dry ready-mix concrete to enhance reproducibility. The concrete, identified in Section 3.1 as a fast-setting mix (QUICKCRETE), was chosen for its availability and ease of use. Section 3.4 details the preparation of the fresh mix, including precise amounts of dry mix and water. While a detailed table of constituent materials was initially considered, the necessary information was not provided on the product packaging. This approach ensures that other researchers can replicate the study using the specified commercial product.
• Table 1: Capital letter in line 3
This has been corrected.
• Chapter 3.2: What was the average size of the fibres and standard deviation?
Section 3.3 specifies the fiber dimensions: length of 48-52 mm, width of 2-3 mm, thickness of 0.2 mm, and an aspect ratio of 250.
• Chapter 3: What about statistical analysis? How many samples were tested in each research?
Sections 3.4 and 3.5 specify that five specimens were used for each test group. Where applicable, plots include error bars indicating the standard error of the mean. The sample groups generally exhibited consistent behavior.
• Chapter 3: Add a table with sample designation. Explain why different mixtures were used for the presented research.
A table with sample designations has been added. It is clarified that the same concrete mix was used across all tests to ensure consistency.
• Chapter 3.8: How were the samples prepared?
Section 3.8 has been rewritten to explicitly detail the sample preparation process.
• Chapter 4.6: Add some figures presenting the SEM images. What was exactly visible under very high resolutions such as 20,000x?
Representative SEM images have been included in Section 3.8 and 4.6. At 20,000x magnification, the images clearly depict solid and void areas. The high resolution allowed for accurate analysis of both solid area calculation and entropy. Images at multiple magnifications (2,500x, 5,000x, 10,000x, and 20,000x) were analyzed to ensure clear visualization of voids, represented by black regions. The solid area and entropy values were averaged across images from the same sample group for robust analysis.

Reviewer 2 Report
Comments and Suggestions for Authors
The manuscript presents research on the possibility of using the special property of chitosan fibers, which is shrinkage in an alkaline environment, to reduce early cracking of cement concrete. The concept is interesting and innovative, although I consider the research itself to be very preliminary.
The research was planned and conducted correctly. Their description is clear, although I have a few comments and questions. The conclusions in the summary correspond to the research results.
I will present the comments in points:
1) The authors used a commercially available product (QUICKCRETE) as a concrete mix. Is there any more detailed information available (mix proportions, type of cement)?
2) Were any mechanical properties of the chitosan fibers determined (or are they available in the literature)?
3) Rectangular molds with a 2 mm extraction hole were used, I do not fully understand what the authors mean by writing about the hole and what its purpose was, maybe it would be worth showing a photo?
4) Why was manual mixing with a scoop used, were any procedures followed (apart from 30 seconds of vibration)
5) Instead of describing the apparatus (lines 261-263) a diagram can be shown in the figure
6) Figure 3 shows that both types of fibers shrink, only with different intensity, therefore it seems to me that in line 322 it would be more appropriate to specify "intensity of shrink" instead of "ability to shrink"
7) Can the distinctive behavior of the mixture with 0.24 wt% content of active fibers be justified or explained in any way? This content is clearly outside the general trend that can be observed in figures 4-6.
8) In my opinion, examples of SEM images should be shown in chapter 4.6 and the methodology for calculating the entropy of the images should be given. I also do not understand the relationship between entropy and the characteristics of concrete.
9) Line 191 - typo error - 2.5kg/m3
Author Response

(The authors gave the same response as above.)

Reviewer 3 Report
Comments and Suggestions for Authors
The submitted manuscript is focused on the enhancement of concrete durability and long-time resilience by incorporating shrinking chitosan-based fibers. The paper is potentially interesting for the readers and with the promising outcomes for the building practice, especially for the design of more durable concrete resistant to chlorides and freeze/thaw cycles. The manuscript could be processed further to publication, but several revisions must be carried out. See my specific comments below.
- In Abstract, there is a typo. It needs to be removed.
- The novelty of the manuscript is not sufficiently elaborated. There are several already published papers aimed at the chitosan fibers used in concrete. These should be referenced and critically evaluated to prove the novelty of the manuscript. See, e.g. https://www.sciencedirect.com/science/article/pii/S0950061823038771 and introduced references.
- What were the real dimensions of the fibers used? What was their aspect ratio?
- Except figure 3, where relative dimensions of the fibers are presented, figure showing the changes of the real dimensions of the fibers might be added.
- Apart from Entropy of SEM images, the structure of the samples, maybe some of them, should be presented in SEM images and the formed hydrated products, aggregates, fibers used and ITZ characterized.
- To characterize the cast and hardened concrete samples, at least bulk density and compressive strength should be added to the manuscript.
- What was the effect of chitosan fibers on the workability of fresh mixtures, e.g. on their setting or spreading.
Author Response
Response to Reviewers' Feedback
We sincerely thank the reviewers for their time, thoughtful insights, and valuable suggestions. Their feedback has been crucial in strengthening the manuscript and enhancing its potential for publication. Below is a detailed, point-by-point response to each of the three reviews.
Reviewer 3:
• In the Abstract, there is a typo. It needs to be removed.
The identified typographical error (double point) has been removed from the abstract.
• The novelty of the manuscript is not sufficiently elaborated. There are several already published papers on the use of chitosan fibers in concrete. These should be referenced and critically evaluated to establish the manuscript's novelty.
The Introduction section has been completely rewritten to provide a more comprehensive overview of existing research on the use of chitosan fibers in concrete. Relevant studies, including the one cited by the reviewer, have been critically reviewed to highlight the current state of knowledge. The manuscript now clearly articulates the unique aspects of this research, particularly its focus on the innovative application of self-shrinking fibers chemically activated by cement reactions. This approach aims to enhance concrete durability by reducing internal pore sizes—a concept that has not been extensively explored in prior studies.
• What were the real dimensions of the fibers used? What was their aspect ratio?
The actual dimensions of the chitosan fibers, including their length of 48-52 mm, width of 2-3 mm, thickness of 0.2 mm, and calculated aspect ratio of 250, have been added to Section 3.3 for clarity and precision.
• Except for Figure 3, where relative dimensions of the fibers are presented, a figure showing the changes in the real dimensions of the fibers might be added.
Figure 3 already effectively illustrates the changes in fiber dimensions at various stages, serving as a visual representation of the corresponding data. Additionally, the underlying table containing detailed measurements of the fibers at each stage has been included for the reviewer's reference. However, we believe that incorporating this table into the main manuscript would not add substantial value, as the information is already effectively conveyed in Figure 3.
Fiber # Dimension Original (mm) After alkaline bath (mm) After drying (mm) After alkaline bath 2 (mm) After drying again (mm)
1 Length 50 50.82 17.44 47.62 14.45
Width 2.5 2.87 1.4 2.5 1.51
2 Length 50 47.74 11.27 41.47 10.37
Width 2.5 3.32 1.69 2.54 1.55
3 Length 50 50.51 13.31 39.94 14
Width 2.5 2.14 1.24 1.91 1.2
4 Length 50 50.81 15.2 48.24 26.16
Width 2.5 3.15 1.83 2.47 1.76
5 Length 50 45.65 14.81 41.58 17.99
Width 2.5 2.41 1.39 1.93 1.51
• Apart from entropy analysis of SEM images, the structure of the samples, including formed hydrated products, aggregates, fibers, and the interfacial transition zone (ITZ), should be characterized.
To enhance the manuscript, two SEM images have been added:
Section 3.8 now includes an image showing the size of the concrete sample prepared for SEM scanning.
Section 4.8 presents an example SEM image illustrating the structure of one of the scanned samples.
Regarding the characterization of formed hydrated products and the ITZ, it is important to clarify that the SEM analysis was conducted at a late stage—specifically on chloride penetration specimens. The primary objective of this scanning was to assess the dimensional changes in the chitosan fibers specimen compared to control specimens. Due to the advanced stage of hydration and the nature of the specimens, the SEM analysis did not yield sufficient data on hydrated products or ITZ characteristics. This limitation has been acknowledged in the manuscript. Future research will aim to conduct earlier-stage scanning for more comprehensive microstructural characterization.
• To characterize the cast and hardened concrete samples, at least bulk density and compressive strength should be added to the manuscript.
• What was the effect of chitosan fibers on the workability of fresh mixtures, such as their setting or spreading behavior?
The primary objective of this research was to investigate the effect of self-shrinking chitosan fibers on the internal pore structure of concrete and to correlate this with durability enhancements. Specifically, the focus was on determining whether fiber-induced shrinkage could lead to smaller pore sizes, resulting in denser microstructures and reduced permeability. Given this focus, measurements of compressive strength, bulk density, or workability parameters (such as setting time or slump) were not within the scope of this preliminary investigation.
Furthermore, this study was designed as a proof-of-concept, aiming to introduce an innovative approach to concrete durability through the use of chemically activated, self-shrinking fibers. Although these additional tests were not conducted, the manuscript clearly specifies the type of dry mix used, the exact water-to-dry-mix ratio, and the detailed mixing procedure. This level of detail is intended to ensure the study's reproducibility.
Nonetheless, we acknowledge the importance of comprehensive material characterization and plan to include bulk density, compressive strength, and workability assessments in future research. These additional studies will build upon the current findings and provide a more holistic evaluation of the proposed fiber-reinforced concrete system.

Round 2
Reviewer 2 Report
Comments and Suggestions for Authors
The manuscript has been significantly rewritten. I thank the authors for the corrections made. Overall, I am satisfied. In the future, I recommend that the authors use their own concrete mixtures - using ready-made products (Quickcrete) without knowing their composition makes it impossible to confirm the results by other researchers. A similar mistake is the lack of clear criteria for mixing the ingredients.
The manuscript can be published as is.
Reviewer 3 Report
Comments and Suggestions for Authors
Based on the revisions made, the manuscript can be accepted for publicaiton in Materials.